# Ruthenium Complexes: An Alternative to Platinum Drugs in Colorectal Cancer Treatment

**DOI:** 10.3390/pharmaceutics13081295

**Published:** 2021-08-19

**Authors:** Kazi Mustafa Mahmud, Mahruba Sultana Niloy, Md Salman Shakil, Md Asiful Islam

**Affiliations:** 1Department of Biochemistry and Molecular Biology, Jahangirnagar University, Savar, Dhaka 1342, Bangladesh; kazi.stu2012@juniv.edu (K.M.M.); mahruba.niloy@gmail.com (M.S.N.); 2Department of Pharmacology & Toxicology, University of Otago, Dunedin 9016, New Zealand; 3Department of Biochemistry, Primeasia University, Banani, Dhaka 1213, Bangladesh; 4Department of Haematology, School of Medical Sciences, Universiti Sains Malaysia, Kubang Kerian 16150, Malaysia

**Keywords:** ruthenium, nanoparticles, colorectal cancer, treatment, diagnosis

## Abstract

Colorectal cancer (CRC) is one of the intimidating causes of death around the world. CRC originated from mutations of tumor suppressor genes, proto-oncogenes and DNA repair genes. Though platinum (Pt)-based anticancer drugs have been widely used in the treatment of cancer, their toxicity and CRC cells’ resistance to Pt drugs has piqued interest in the search for alternative metal-based drugs. Ruthenium (Ru)-based compounds displayed promising anticancer activity due to their unique chemical properties. Ru-complexes are reported to exert their anticancer activities in CRC cells by regulating different cell signaling pathways that are either directly or indirectly associated with cell growth, division, proliferation, and migration. Additionally, some Ru-based drug candidates showed higher potency compared to commercially available Pt-based anticancer drugs in CRC cell line models. Meanwhile Ru nanoparticles coupled with photosensitizers or anticancer agents have also shown theranostic potential towards CRC. Ru-nanoformulations improve drug efficacy, targeted drug delivery, immune activation, and biocompatibility, and therefore may be capable of overcoming some of the existing chemotherapeutic limitations. Among the potential Ru-based compounds, only Ru (III)-based drug NKP-1339 has undergone phase-Ib clinical trials in CRC treatment.

## 1. Introduction

Colorectal cancer (CRC) is a type of malignant neoplasm of the colon or rectum epithelial cell lining [1,2], which is recognized as the third most prevalent cancer worldwide and is the fourth leading cause of death [3,4]. It also accounts for about 10% of all yearly diagnosed cancers and cancer-related deaths globally [5]. Moreover, CRC has been documented as the second and third most common cancer in women and men, respectively [5,6]. CRC occurrence rate is high in most of the developed countries, whereas the rate is increasing rapidly in developing countries [6]. In 2020, more than 1.9 million individuals were estimated to be diagnosed, where 935,000 individuals would die among the CRC-diagnosed patients [7]. About 2.5 million people are predicted to be diagnosed with CRC by 2035 [5].

The conventional treatment strategies of CRC consist of surgical resection, radiation, and chemotherapy, which may extend the survival rate by only five years in 90% of stage I patients to 10% of stage IV patients [8,9]. Even though surgery has been an integral part of CRC treatment, it comes out with post-operative complications such as occurrence or acceleration in recurrence of tumor cells and/or development of liver metastasis [8]. Long-term use of chemotherapeutics and radiation induces peripheral neuropathy [10] and bowel dysfunction accompanied by increased frequency and urgency problems [11]. The limitations of the existing treatment strategies encourage researchers to develop effective therapeutic alternatives.

Over the past few decades, transition metal-based compounds have been extensively used in the anticancer medicinal chemistry area [12,13,14,15]. Platinum (Pt)-based medications such as cisplatin (CIS) and its analogs carboplatin (CAR) and oxaliplatin (OXA) (Figure 1) have been used worldwide in cancer treatment [16]. Additionally, some other Pt-based drugs, for example, miriplatin (Japan), nedaplatin (Japan), lobaplatin (China), and heptaplatin (Korea) (Figure 1) are used regionally in cancer treatment (Figure 1) [17]. However, only OXA has been approved by the Food and Drug Administration (FDA) in CRC treatment [18] and stands out as the first-line therapy against CRC [19]. Despite being highly efficient, OXA has severe side effects [20] and drug resistance [21]. Such limitations inspire the search for alternative metal-based anticancer drugs.

Among other transition metals, ruthenium (Ru) is a better alternative to Pt [22]. Ru displays both early and late transition metal properties due to its central position in the second row of the transition metal series [22]. The 4d subshell of Ru is partially filled and it contains many valencies that enable Ru to form a wide range of complexes via π bond formation, which can perform as anticancer agents against various tumor cell lines [23]. Ru-complexes showed promising anti-proliferative activity in vitro, in vivo, and in chemical model systems [24,25,26]. Moreover, the Ru-complex showed synergistic activity when combined with established anticancer agents and drugs [25,27]. Furthermore, Ru-complexes are widely used as phototherapeutic agents, biomolecular probes, and bioimaging reagents [28]. Luminescent Ru-complexes can differentiate DNA structures and have the potential to be used as molecular light switches for DNA [29]. Additionally, Ru nanoparticles (RuNPs) can be used as a cancer theranostic agent for the early diagnosis and treatment of CRC [30,31]. Nanostructured Ru-complexes offer improved anticancer activity under their targeted drug delivery and reduced side effects [32].

To overcome the limitation of Pt-drugs, Ru-complexes could be used as an alternative to Pt-based chemotherapeutic drugs in CRC. In this review, we scrutinized the potential of ruthenium-based drugs, drug candidates, and ruthenium nanoparticles in the treatment of CRC. Additionally, the molecular mechanism of action(s) such as effects on nucleic acids, cell proliferative pathways, and cell cycle are summarized and compared their efficiency with Pt-based drugs and other chemotherapeutic drugs i.e., 5-Fluorouracil (5-FLU), Doxorubicin (DOX), and Etoposide (ETP).

## 2. Colorectal Cancer and Pt-Based Drugs

Colorectal cancer is caused by chromosomal instability, microsatellite instability (MSI), and the CpG island methylator phenotype (CIMP), which may occur alone or in combination [33,34]. Chromosomal instability is responsible for most of the genetic instability in CRC, which is characterized by significant gain or loss of entire or large portions of chromosomes [33]. The chromosomal instability pathway starts with the mutation of the *APC* gene, followed by the mutation of oncogene KRAS and inactivation of tumor suppressor gene, TP53 [35]. The CIMP pathway is involved in hypermethylation of the promoter region of tumor suppressor genes, mostly MGMT and MLH1. However, this hypermethylation is linked to BRAF mutation and MSI [36]. The MSI pathway refers to the inactivation of DNA mismatch repair genes through genetic alteration in short repeated sequences and hypermethylation of these mismatch repair genes. The MSI pathway is often found to be connected to the CIMP pathway [34].

Several genes have been mutated to induce CRC; Figure 2 shows the ten most frequent genes according to the cBioPortal database (https://www.cbioportal.org/, accessed on 15 July 2021) [37,38] that calculated published data on CRC [39,40,41,42,43,44,45]. Mutations of these genes could be linked with survival, CRC progression, and therapeutic outcome. 

Pt-based drugs are used in the treatment of various types of cancers including CRC [47]. Although Pt-based drugs have been playing a pivotal role as anticancer drugs, some irresistible drawbacks limit their use in cancer treatment. Like other conventional chemotherapeutic drugs, Pt-based drugs including CIS, OXA, and CAR display poor cancer cells’ selectivity index [48,49,50]. Due to low selectivity, patients often experience drug-induced complications, some of which are fatal [51]. 

Among the Pt-based drugs, only OXA has been used in the treatment of CRC. OXA in combination with leucovorin (LEU) and 5-FLU (FOLFOX) is administered in adjuvant or neoadjuvant treatment of CRC. However, the co-treatment increases all grades of anemia significantly compared to the individual treatment with LEU and 5-FLU [52]. Furthermore, OXA induced several side effects in CRC treatment including peripheral neuropathy, fatigue, diarrhea, nausea, and stomatitis [53,54]. Acute and chronic neurosensory symptoms are also observed after OXA treatment [55]. Additionally, OXA mediates neutropenia, the most common serious hematological toxicity, in CRC patients [53]. Besides, continued use of OXA develops hypersensitivity reactions (type-I or IgE mediated reactions) in 10% of patients which is characterized by pruritus, flushing, urticarial, hypotension, and possible angioedema of the larynx, face, and/or extremities [54]. Moreover, OXA is reported to induce hepatic sinusoidal injury in CRC patients [56] as well as enlargement of spleen size in stage II or III CRC which are the potential cause of persistent thrombocytopenia [57].

## 3. Features of Ru-Complexes

Among numerous transition elements, Ru is found to be the best alternative to Pt [12,22,58]. The advantages of using Ru over Pt include lower toxicity, a broader range of oxidation states (2^+^, 3^+^, and 4^+^), a slow rate of ligand exchange, and the ability to mimic iron that facilitates its binding to human serum transferrin and other proteins [59,60]. Ru offers octahedral coordination geometry instead of square-planar geometry of Pt(II) complexes which provide a different mode of action and reactivity than CIS [61]. Furthermore, compared to typical Pt-based drugs, many Ru-based compounds have better water solubility in the biological environment, resulting in improved effectiveness against Pt-drug resistant tumor cells [62]. This increase in water solubility may aid in balancing the hydrophilicity and hydrophobicity of Ru-complexes, resulting in increased absorption in cancer cells [63,64].

Ru(IV) is unstable because of the higher oxidation state. This limits the antitumor effects and further development of Ru(IV)-complexes [65]. Nevertheless, Ru(II) and Ru(III) have antitumor activity [66]. Ru(III)-complexes possess stable thermodynamics and kinetics and are efficient in acting as a prodrug to work under hypoxic and acidic conditions [67]. However, Ru(III) is considered to be more inert than Ru(II), which might be due to a higher effective nuclear charge [68]. Thus, Ru(II)-complexes are more reactive than Ru(III)-complexes [69]. Ru(III)-complexes are reduced to the more active form, Ru(II), by the “activation by reduction” mechanism [70]. This mechanism is influenced by cellular reducing agents such as ascorbate, glutathione, and hypoxic tumor microenvironment [23,71]. Reduction of Ru(III) to Ru(II) enervates π bond with donor ligand and elevates ligand substitution rates [23]. However, the “activation by reduction” hypothesis is still a controversial issue, as some Ru(III)-complexes remained at 3^+^ oxidation state after 24 h of intravenous administration [72].

## 4. Underlying Mechanisms of Ru-Complexes in Targeting CRC 

Different types of Ru-complexes are reported to target DNA, different fundamental enzymes like topoisomerase II, thioredoxin reductase, and various biomolecules linked with growth, angiogenesis, migration, metastasis, and apoptosis of CRC cells (Figure 3) [27,73,74,75,76,77,78,79,80,81,82,83,84,85,86,87,88]. Furthermore, Ru-complexes induce apoptosis through overproducing reactive oxygen species (ROS) [89] and compromising cellular organelles that are required for cell survival [90,91]. Moreover, Ru-complexes also cause apoptosis in CRC cells through photodynamic activity [92]. In this section, we will compare the potency of some Ru-based complexes with Pt-drugs along with another standard drug 5-FLU, and investigate the mechanism of action(s), and promise of Ru-based complexes in CRC treatment. Herein, we considered comparing 5-FLU alongside Pt-based drugs since it has been used as a first-line treatment against CRC [93].

### 4.1. DNA Damage Mediated Apoptosis

Many Ru-complexes control the proliferation of cancer cells through DNA damage [94,95]. Some of the Ru-complexes bind with DNA via electrostatic attraction, major or minor grooves binding, intercalative binding mode, or by a combination of these two or more [96,97]. DNA binding modes of Ru-complexes can be confirmed by UV-Vis spectroscopy, viscosity investigation, and fluorescence spectrometry [97]. Intercalative binding opens a gap between the DNA base pair and inserts planar aromatic molecule of anticancer drug above and below the bases [98]. This results in unwinding and lengthening of the helix structure of DNA (Figure 3→I) [97,99]. Ru(II)-complexes bind preferentially to N7 of guanosine and N3 of thymidine, but insubstantially to N3 of cytidine, and little to adenosine [100]. On the other hand, Ru(III)-complexes preferentially bind to phosphate groups of the DNA backbone. This is attributed to the strong electrostatic interaction between tricationic Ru(III) fragment and anionic phosphate groups [101]. Ru(III)-complexes also have been reported to bind to the N7 site of guanine [102] but in a less pronounced way than phosphate [101]. However, both Ru(II) and (III)-complexes have a common tendency to bind to the N7 site of guanine which is similar to CIS [103].

Some Ru-complexes such as Ru(III)-PTA compound trans-[RuCl_4_(1,3,5-triaza-7-phosphaadamantane protonated at one N atom)_2_]Cl (**1a**), η6-arene Ru-complexes particularly Ru-5-chloro-3-((5-(3-(4-methyl-1,4-diazepane-1-carbonyl)phenyl)furan-2-yl)methylene)indolin-2-one (**1b**), Ru(II) arene complexes including [(η6-fluorene)RuII(ethylenediamine)Cl]^+^ (**1c**), and [(η6-9,10-dihydrophenanthrene)RuII(ethylenediamine)Cl]^+^ (**1d**) mediated apoptosis in CRC cells by damaging DNA (Figure 4) [75,76,77]. **1b**, **1c**, and **1d** bound with guanine residue of DNA, causing DNA fragmentation [76,77]. Though the compounds share the same mechanism of action(s), they are not equally potent towards different CRC cells in terms of IC_5O_ values (Table 1). **1a** with an IC_50_ value of ˃100 µM displayed moderate anti-proliferative activity against HCT116 cells following 24 h of treatment [75]. Likewise, **1b** showed cytotoxicity in LoVo cells (IC_50_ = 8.1 µmol/L) and LS174T (IC_50_ = 7.7 µmol/L) after six days of treatment [76]. However, both **1a** and **1b** were less potent compared to CIS (Table 1) [75,76]. In contrast, **1c** and **1d** exhibited synergistic action with ionizing radiation (IR) and were more potent than OXA against DLD1 cells (Table 1) [77]. OXA act as an alkylating agent on DNA, forming intra-strand cross-links between two adjacent guanine or two adjacent guanine-adenine and inhibit DNA synthesis through disrupting replication and transcription [104,105]. Like OXA, **1c** or **1d** form adducts with DNA which leads to apoptosis. Experimental results indicate that **1c** or **1d** was more potent than OXA towards CRC cell lines, therefore these drug candidates could be used as an alternative to OXA.

### 4.2. Inhibition of Topoisomerase II Enzyme

Topoisomerase II is involved in modulating topological problems related to DNA replication, transcription, chromatin remodeling, and recombination by single- or double-strand breaks in the DNA [106]. Half-sandwich Ru-arene complexes with thiosemicarbazones including [(η6-p-cymene)Ru(piperonal-*N*(4)-ethylthiosemicarbazone)Cl]Cl (**2a**), [(η6-p-cymene)Ru(piperonal-*N*(4)-phenylthiosemicarbazone)Cl]Cl (**2b**), and mixed-ligand diimine-piperonal thiosemicarbazone complexes of Ru(II) [(1,10-phenanthroline)2Ru(2-(benzo[d][1,3]dioxol-5-ylmethylene)-*N*-methylhydrazinecarbothioamide)](hexafluorophosphate)2 (**2c**) (Figure 4) act as topoisomerase II inhibitor by disrupting enzyme’s catalytic cycle which leads to apoptosis of CRC cells by inhibiting replication (Figure 3→II) [73,74]. Both **2a** and **2b** exhibited less anti-proliferative activity compared to CIS and ETP [73] while **2c** was more potent than ETP against HCT116 and Caco-2 cells (Table 1) [74]. Although ETP has been widely used as an anti-proliferative agent [107], it is ineffective against advanced CRC [108,109]. The combination of CIS and ETP showed low anticancer activity against advanced CRC [110]. Thus, it is not prudent to use only EPT while comparing anticancer activity against CRC.

### 4.3. MAPK Signaling Pathway

Mitogen-activated protein kinase (MAPK) signaling pathway regulates apoptosis in CRC cells through controlling three MAPK family proteins—extracellular signal-regulated kinase (ERK) [111], c-Jun N-terminal kinases (JNK/SAPK) [112] and p38 MAPK [113]. Some Ru-complexes including Ru(II) naphthalimide N-Heterocyclic Carbene compounds (**3a**), [Ru(piplartine)(1,1-bis(diphenylphosphino)ferrocene)(2,2′-bipyridine)](hexafluorophosphate)2 (**3b**), [Ru(piplartine)(1,4-bis(diphenylphosphino)butane)(2,2′-bipyridine)](hexafluorophosphate)2 (**3c**), and Ru(II)-thymine complex [Ru(triphenylphosphine)2(thyminate)(2,2′-bipyridine)]hexafluorophosphate (**3d**) (Figure 4) triggered apoptosis in CRC cells through MAPK (JNK, p38 MAPK, and ERK1/2) signaling pathway (Figure 3→III) [79,80,81]. 24-48 h post-treatment of HCT116 cells with **3a** (12 µM), **3b** (2.5 µM), **3c** (5 µM), and **3d** (4 µM) induced apoptosis of HCT116 cells by controlling MAPK (JNK, p38 MAPK, and ERK1/2) signaling pathways [79,80,81]. Furthermore, **3b** (2.5 µM), **3c** (5 µM), and **3d** (4 µM) increased apoptosis of HCT116 cells by 19%, 23%, and 51%, respectively compared to chemotherapeutic drug DOX (1 µM). Afterward, in vivo study showed that intraperitoneal injections of **3b** (15 µmol/kg/day), **3c** (15 µmol/kg/day), and **3d** (1–2 mg/kg/day) for 15 consecutive days reduced tumor mass weight by 1.55, 1.42, and 1.47 to 1.67-fold, respectively compared to negative control in C.B-17 SCID mice model engrafted with HCT116 cells [80,81]. It should be noted that **3d** (1–2 mg/kg/day) displayed 32.6–40.1% tumor mass inhibition rate while established drug 5-FLU (15 mg/kg/day) showed 62.7% [81]. Since **3d** showed higher potency compared to 5-FLU, thus **3d** could be used as an alternative to 5-FLU in KRAS-mutated CRC treatment.

**Figure 4 pharmaceutics-13-01295-f004:**
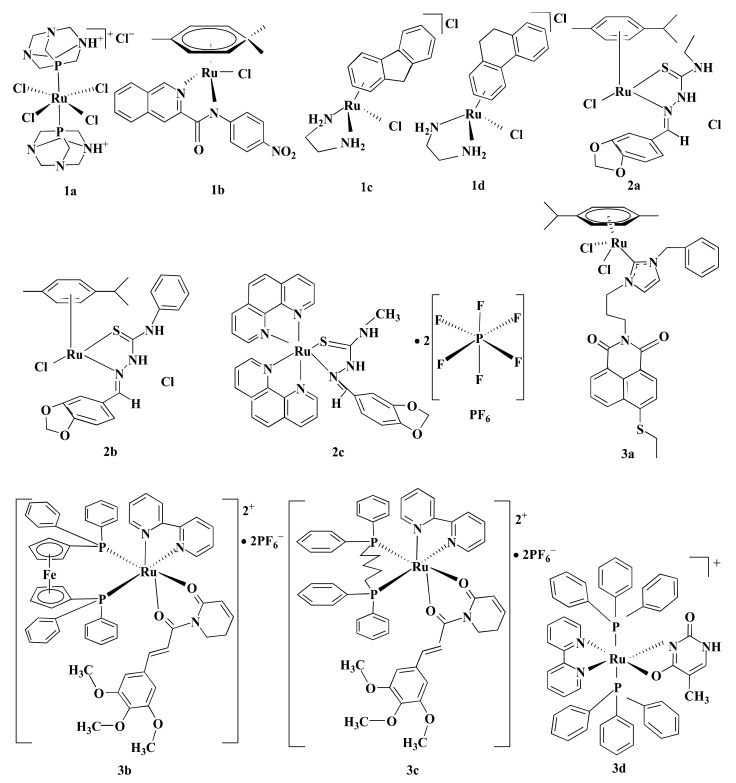
Schematic representation of some Ru-based drug candidates. Different Ru-complexes mediated CRC cells apoptosis through DNA damage (**1a**–**d**), inhibiting DNA topoisomerase II enzyme (**2a**–**c**), and regulating MAPK signaling pathway (**3a**–**d**).

### 4.4. p53 Dependent Caspase-3 Mediated Signaling

p53 is a tumor suppressor protein linked with cell cycle, apoptosis, senescence, and autophagy [114]. Several Ru-complexes including Ru-based 5-FLU complex [Ru(5-fluorouracil)(triphenylphosphine)_2_(2,2′-bipyridine)]hexafluorophosphate (**4a**), Ru-Quercetin (**4b**), Ru-Phloretin (**4c**), Ru-Baicalein (**4d**), and [Ru(biphenyl)Cl(1,2-ethylenediamine)]^+^ with hexafluorophosphate (**4e**) (Figure 5) induced apoptosis of colon cancer cells via p53 dependent caspase-3 mediated apoptosis (Figure 3→IV) [27,82,83,84,85]. **4b**, **4c**, and **4d** upregulated the expression of p53, and anti-apoptotic protein Bax while downregulated Bcl-2 expression, resulting in activation of apoptotic protein caspase-3 to induce apoptosis and arrested HT29 cells at G0/G1 phase in a concentration-dependent manner [82,83,84]. Moreover, these compounds also displayed apoptotic activity in the 1,2-dimethylhydrazine (DMH) and dextran sulfate sodium (DSS) induced CRC in male Wistar rats [82,83] and Swiss albino mice [84]. Treatment with **4b** (200 mg/kg) increased the expressions of p53 (3.47-fold) and Bax (3.77-fold) while suppressing the expression of Bcl-2 (2.72-fold) compared to the DMH and DSS treated group [83]. Similarly, the same concentration of **4d** and **4c** upregulated Bax expression (2.45 to 2.6-fold) while downregulated Bcl-2 expression (2.38 to 3.47-fold) compared to the DMH and DSS treated group. The upregulation and downregulation of proteins (i.e., p53, Bax, and Bcl-2) expression were determined by immunohistochemical analysis [82,84]. Additionally, **4b**, **4c**, and **4d** diminished the cellular level of proliferating cell nuclear antigen (PCNA) which is actively controlled by p53 and subsequently induced apoptosis by minimizing cell proliferation [82,83,84].

Treating HCT116 cells with **4e** (15–60 µM) for 48 h increased the expression of p53, Bax and cell cycle inhibitor p21/WAF1 (Table 2). Protein expression levels were determined by western blotting. However, p53/Bax-null HCT116 cells did not undergo significant apoptosis in the same condition. This implies the importance of p53 and Bax proteins in **4e** mediated apoptosis of CRC cells. Furthermore, treatment with **4e** brought about a long-term loss of cellular replication in a p53, Bax, and p21/WAF1 independent manner [85]. Silva et al. [27] developed a novel Ru-based complex **4a**, where the addition of Ru increased the cytotoxic potential of 5-FLU. **4a** was more potent than 5-FLU (2.73-fold) and OXA (2.87) while less potent than DOX (3-fold) (Table 1). Furthermore, trypan blue exclusion (TBE) assay pointed that treatment of HCT116 cells with **4a** (4 µM) for 48 h increased caspase-3 level (3 and 3.75-fold) along with mitochondrial membrane depolarization (2.2 and 2.29-fold) compared to OXA (2.5 µM) and 5-FLU (4 µM), respectively [27]. Another Ru-based complex [(η^5^-C_5_H_5_)Ru(1,2-Bis(diphenylphosphino)ethane)]^+^ bearing a galactose ligands (**4f**) (Figure 5) also triggered caspase-3 and caspase-7 activities level and thus induced apoptosis. A study by Florindo et al. [115] reported that treatment of HCT116 cells with **4f** (1–2 µM) reduced cell viability (40% to 16%) and increased cell death (1.5 to 2.7-fold) compared to control. Though **4f** was equally potent to OXA with an IC_50_ value of 0.45 µM in HCT116 cells. However, at 2 µM concentration, **4f** was 25% more cytotoxic than OXA. **4f** (0.45 µM) significantly increased caspase-3 and caspase-7 activity by 1.4-fold than control, while OXA (0.45 µM) increased the level of caspase-3 and caspase-7 by 1.27-fold. Besides, **4f** induced 30% more apoptosis in HCT116 cells than OXA at an equal concentration of 2 µM [115]. Based on the IC_50_ values, **4a** and **4f** were more or equally potent compared to 5-FLU or OXA against HCT116 cells. Therefore, **4a** and **4f** could be used in CRC treatment in place of 5-FLU and OXA.

### 4.5. Upregulation of APC and p53 Gene

APC and p53 genes are reported to acquire genetic alteration in CRC [121]. About 80% APC mutation and 60% p53 mutation are observed in CRC [122,123]. APC performs antitumorigenic activity by regulating β-catenin levels [123]. Mutation of APC causes accretion of β-catenin which in turn translocates to the nucleus and influences transcription factor Tcf/Lef to transcribe cyclin D1, C-Myc, and CRD-RB that play role in cell-cycle progression, growth, and proliferation [124]. p53 regulates genes associated with DNA repair, cell cycle arrest, and apoptosis [125]. Some Ru-based metalla-bowl compounds showed anticancer activity among which [Ru_4_(p-cymene)_4_-(5,8-dioxydo-1,4-naphthaquinonato-)_2_(2,6-bis(N-(4-pyridyl)carbamoyl)pyridine)_2_][4CF_3_SO_3_] (**5a**) (Figure 5) was most potent and mediated CRC cells apoptosis by upregulating APC and p53 genes expression (Figure 3→V). After 24 h, **5a** exhibited higher anti-proliferative activity against HT-15 cells compared to CIS and DOX (Table 1) [86]. Additionally, **5a** (2 µM) upregulated the expression of APC mRNA (2.9-fold) and p53 mRNA (4.1-fold) in HCT116 cells compared to the untreated control group [86]. Results indicate that **5a** displayed anti-proliferative activity via the upregulation of APC and p53 genes expression. As the tested control drugs, DOX and CIS are not used in CRC treatment, experimentation using OXA or 5-FLU could be used in further studies.

### 4.6. p53 Independent Activity 

Ru-complexes including [Ru(η^6^-*p*-cym)(7-(4-(Decanoyl)piperazin-1-yl)-ciprofloxacin_-H_)Cl] (**6a**) and Ru-arene Schiff-base complexes particularly [(η6-1,3,5-triisopropylbenzene)RuCl(4-methoxy-*N*(2-quinolinylmethylene)aniline)]Cl (**6b**) (Figure 5) exhibited p53 independent anticancer property against CRC cells that are resistant to Pt-based anticancer drugs [116,117]. CRC cells overexpress organic cation transporter (OCT) proteins that are responsible for resistance towards CIS. Conversely, OXA contains hydrophobic 1,2-diaminocyclohexane (DACH), which produces cationic species. These cationic species act as OCT1/2 substrates [126]. Similarly, **6b** bear hydrophobic ligands and are constantly cationic. Therefore, it acts as potential substrates of OCT1/2 and is effective against CRC cells [117]. It was evident that **6a** and **6b** were more potent compared to CIS and OXA against HCT116 cells (Table 1) [116,117]. Furthermore, **6a** was reported to arrest the S phase followed by the G2/M phase of cell cycle while **6b** arrested cell cycle at both G0/G1 (2.5 µM) and G2/M (10 µM) phase in a dose-dependent manner [116,117]. Considering their IC_50_ values, **6b** was more potent than OXA against HCT116 cells and could be used as an alternative to OXA in treating CRC with dysfunctional p53. 

### 4.7. Inhibition of Proteasome

Proteasome activity is crucial for various cellular processes like cell-cycle regulation, cell differentiation, angiogenesis, and apoptosis [127,128,129]. Thus, inhibition of proteasome activity is a potential way to mediate apoptosis in cancer cells [130]. Curcumin, a biocompatible compound is reported to induce apoptosis in CRC cells through inhibiting proteasome activity [131]. Ru(II) arene complexes with curcumin including [Ru(p-cymene)(curcumin)Cl] (**7a**), [(Benzene)Ru(curcumin)Cl] (**7b**), and [Ru(hexamethylbenzene)(curcumin)Cl] (**7c**) (Figure 6) induced DNA fragmentation leading to apoptosis via proteasome inhibition in CRC cells [90]. Generally, p53 and Poly-(ADP)-ribose polymerase (PARP) increases upon DNA damage and are engaged in repairing damaged DNA [132,133]. However, extensive fragmentation of DNA leads caspase-3 to cleave 116 kDa PARP into 85 kDa fragment, resulting in the inactivation of ubiquitin protease function and triggering apoptosis (Figure 3→VII) [90,133]. Among three curcumin-based Ru-complexes, **7b** was most potent. Treatment of HCT116 cells with **7b** (10 µM) decreased cell viability (20%–30%) after 4–24 h and reduced 26S proteasome ChT-L activity (37%–25%) after 4 h. The same concentration of **7b** complex also elevated p53 (3.2-fold), caspase-3 activity (25%) while decreased PARP level (2-fold) and 26S proteasomes activities (~30%) compared to control after 24 h in HCT116 cells (Table 2) [90]. The obtained results showed that **7b** inhibited CRC cells proliferation by reducing proteasome activity. As Bonfili et al. did not use any control drug(s) (e.g., OXA), further studies are required to compare the potency of **7b** with standard drug(s). 

### 4.8. ROS-Mediated Apoptosis

ROS are chemically reactive derivatives of oxygen that can exist independently carrying an unpaired electron and show conducive or detrimental effects depending upon its cellular concentration [134,135]. Cellular ROS levels are linked with cancer cell apoptosis, and excessive levels are reported to have anticancer activity [136]. Different Ru-complexes have been reported to modulate apoptosis in CRC cells by elevation of ROS level. Half sandwich Ru(II) complexes [η6-p-cymene)RuCl(2-(5,6-dichloro-1H-benzo[d]imidazole-2-yl)quinolone)] (**8a**) (Figure 6) produced ROS, which in turn damaged DNA via oxidative stress and induced apoptosis in CRC cells (Figure 3→VIII). Considering their IC_50_ values, **8a** displayed 2.90-fold better anti-proliferative activity against Caco-2 cells compared to CIS (Table 1) [118]. Treatment of HT29 cells for 24 h with **8a** (5–6.8 µM) increased ROS level (around 20%–65%) as well as arrest cell cycle more than 65% of cells at G0/G1 phase [118].

Oxidative stress plays a central role in Ru(III)-based drug trans-[tetrachloro-bis(1H-indazole)ruthenate(III)] or KP-1019 (**8b)** (Figure 6) mediated apoptosis in CRC cells. Kapitza et al. [119] reported that **8b** induced H_2_O_2_ formation in CRC cells which further reacts with mitochondrial membrane-embedded unsaturated fatty acids to induce depolarization of the mitochondrial membrane and mediated caspase-3 dependent PARP cleavage. **8b** induced cytotoxicity in both SW480 and LT97 cells with an IC_50_ value of 30 and 50 µM, respectively. In contrast, antioxidant *N*-acetylcysteine (NAC) (5 µM) decreased their potency, as evident from the increase of IC_50_ values to 55 and 88 µM towards SW480 and LT97, respectively. This finding confirmed that ROS is involved in **8b**-mediated apoptosis [119]. In vivo activity of **8b** was evaluated by the chemoresistant MAC15A colon carcinoma in a rat model, closely similar to human colon cancer. Treatment of **8b** (13 mg/kg) two times a week for 10 weeks reduced 8% of tumor size, where 5-FLU (40 mg/kg) reduces tumor size down to only 40% and at the same time, another Pt-based established drug CIS did not show any activity [137,138].

**Figure 6 pharmaceutics-13-01295-f006:**
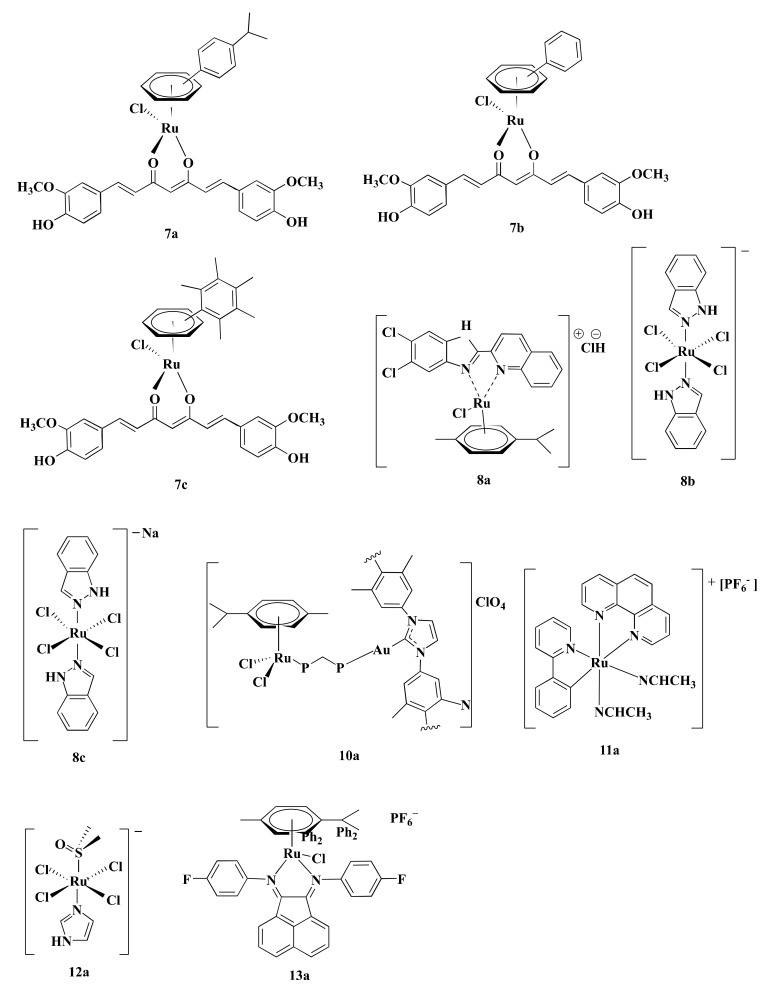
Structures of some Ru-complexes using in CRC treatment. Different Ru-complexes mediated apoptosis of CRC cells via inhibiting proteasome activity (**7a**–**c**), ROS-mediated apoptosis (**8a**–**c**), inhibiting thioredoxin reductase activity (**10a**), inhibiting HIF-1 pathway (**11a**), anti-metastasis activity (**12a**), and lysosomal dysfunction (**13a**).

Sodium salt of **8b**, i.e., sodium trans-[tetrachloride-bis(1H-indazole)ruthenate(III)] or NKP-1339 (**8c**) (Figure 6) was prepared by Keppler et al. [139]. **8c** was reported to increase the ROS level and, therefore, induced ER stress-mediated apoptosis in CRC cells (Figure 3→VIII) [89]. Exposure of **8c** (200 μM) to HCT116 and SW480 CRC cells elevated ROS concentration by 2-fold and 2.5-fold, respectively, after 1 h compared to control and led to apoptosis through ER stress [89]. ROS causes potential damage to proteins that piled up in the ER. Since cancer cells tend to demonstrate an increased level of oxidative stress and ER stress due to having enhanced and fast metabolic activity, hence excessively accumulated misfolded proteins led the ER to start unfolded protein response (UPR) which induced apoptosis after exceeding a certain threshold level (Figure 3→VIII) [89,140]. The underlying mechanism is associated with three transmembrane receptors namely, PrKr-like ER kinase (PERK), activating transcription factor 6 (ATF6), and inositol-requiring protein 1α (IRE1α) which are bound by an ER-resident chaperone, glucose-regulated protein (GRP78), which has high affinity towards misfolded protein [89,140,141]. Upon release from GRP78, PERK phosphorylated eukaryotic translation initiation factor 2α (eIF2α), which increased the cap-independent translation of activating transcription factor 4 (ATF4) [140]. ATF4 consequently translocated to the nucleus and induced transcription factor C/eBP homologous protein (CHOP), which is involved in apoptosis (Figure 3→VIII) (Table 2) [141]. Besides, treatment of both HCT116 and SW480 with **8c** (200 μM) for 6 h mediated translocation of transcription factor Nrf2 from the cytoplasm into the nucleus, which induced different genes containing an antioxidant response element (ARE) in their promoter site to exert antioxidant response [89]. The GRP78 chaperone was found to be regulated on the protein level but only had a slight influence on the mRNA level recommending involvement of ER-associated protein degradation (ERAD) in the mode of action of **8c** [89,141]. Considering all observations, it can be said that ROS plays a vital role in inducing apoptosis in the CRC cells. Among three complexes (**8a**–**c**), **8a** can be considered as the most potent in terms of IC_50_ values. However, experimentation using standard chemotherapeutics drug(s) (e.g., OXA, 5-FLU) could make the study more significant.

### 4.9. Immunogenic Cell Death

**8c** is responsible to mediate ER stress that induces a cascade of events leading to CRC cells death along with providing critical signals to visualize dying cancer cells to the immune system. This consequently introduces sustained immune response against the CRC; a phenomenon termed immunogenic cell death (ICD) [120]. ICD is characterized by secretion of immune-modulatory damage-associated molecular patterns (DAMPs), such as pre-apoptotic calreticulin (CRT) surface-exposure, extracellular adenosine triphosphate (ATP), and high mobility group box 1 (HMGB-1) [142,143]. The ER stress triggers a cascade of reactions that activate PERK. This activated PERK phosphorylates eIF2α which in turn translocate CRT to the cell membrane that is generally located at the lumen of the endoplasmic reticulum of colon cancer cells [144]. Exposure of CRT on cell membrane elicits an “eat me” signal which induces maturation of phagocyte dendritic cell (DC) as well as uptake tumor antigens (Figure 3→IX) [145,146,147]. The other DAMP, HMGB-1 protein, residing in the nucleus moves to the extracellular space in the course of ICD and attach to pattern recognition receptors (PRRs) such as toll-like receptor 4 (TLR-4), receptor for advanced glycation end products (RAGE), and nuclear factor-κB (NF-κB) of DCs and presented antigens from dying tumor cells. This also accelerates DC maturation and migration. HMGB-1 functions as a crucial DAMP showing immune-stimulatory and pro-inflammatory effects [147,148] Lastly, extracellular adenosine triphosphate (ATP), released from dying tumor cell expresses a “find me” signal [147]. Released ATP binds to the purinergic P2RX7 receptors of dendritic cells and activated the (NOD)-like receptor protein 3 (NLRP3) inflammasome. This, in turn, stimulates tumor-specific cytotoxic T cells to secrete IFN-γ [120,147]. IFN-γ have pro-apoptotic and anti-proliferative functions such as inhibiting tumor angiogenesis, inducing regulatory T-cell apoptosis, and influencing M1 pro-inflammatory macrophages activity to suppress tumor progression [149].

Wernitznig et al. [120] described that treatment with **8c** (100 µM) for 24 h upsurged CRT expression by approximately 7%, 10%, and 7% as well as extracellular ATP level by around 3%, 2.5%, and 2.6% in HCT116, HT15, and HT29 cell membrane, respectively, compared to the control. However, **8c** increased the CRT level by 3.75 and 1.25-fold compared to CIS and OXA, respectively, as well as ATP level than both CIS and OXA by 1.36-fold in HT29 cells. Furthermore, the same concentration in HCT116 enhanced the release of HMGB-1 into the cytoplasm (Table 2) [120]. These findings consolidate that **8c** could inhibit CRC cells proliferation via immunogenic death as well as ROS-mediated apoptosis and could be used as an alternative to OXA. 

### 4.10. Inhibition of Thioredoxin Reductase Activity 

Thioredoxin reductase (TrRx) is associated with redox-regulation and cell signaling [150]. Overexpression of TrRx is observed in CRC cells [151,152] and is considered to play a role in resisting CIS [78]. Heterobimetallic Ru(II)–gold(I) complexes [Ru(p-cymene)Cl2(µ-1,1-bis(diphenylphosphino)methane)Au(IMes)]-ClO_4_ (**10a**) (Figure 6) inhibited TrRx activity. **10a** (IC_50_ = 5.22 µM) inhibited TrRx activity of HCT116 cells after 72 h where CIS was unable to induce any effect [78]. As CIS is not an approved drug in CRC treatment, studies using OXA would better reflect experimental findings.

### 4.11. Inhibition of HIF-1 Pathway

Hypoxia-inducible factor-1 (HIF-1) plays a fundamental role in tumor growth, angiogenesis, survival, and energy metabolism [153,154]. Decreased expression of HIF-1 protein level affects in downregulating HIF-1 target genes including vascular endothelial growth factor (VEGF) which modulate tumor angiogenesis [153,155], glucose transporter 1 (GLUT1) that mediate glucose uptake [156], and alpha-enolase (ENO1) which is crucial for glucose metabolism acting as a key catalyzing enzyme in the glycolysis (Figure 3→XI) [157]. All of these genes are responsible for the progression of CRC [158,159,160]. The HIF-1 pathway is regulated by the redox enzyme prolyl hydroxylase 2 (PHD2) [161]. Ru derived compound 11 (**11a**) (Figure 6) was synthesized by Leyva et al. [162]. Vidimar et al. [87], reported that **11a** showed higher anti-proliferative activity than CIS against HCT116 cells (2.7-fold) (Table 1). **11a** interfered with the HIF1 pathway by upregulating PHD2, which consequently decreased the HIF-1α protein level. This, in turn, caused reduced angiogenesis and altered glucose metabolism in CRC cells by decreasing VEGF expression, GLUT1 and ENO1 (Figure 3→XI). Western blotting showed that exposure of HCT116 cells to **11a** (5µM) under hypoxic conditions (1% O_2_) elevated the level of PHD2 enzyme after 6 h. Additionally, treatment of HCT116 cells with **11a** at the same concentration and under the same experimental conditions reduced the expression of VEGF RNA, GLUT1 RNA, and ENO1 RNA by 240%, 320%, and 70%, respectively, while CIS reduced the expression of these genes by 150%, 350%, and 10%, respectively. This reflected better therapeutic efficiency of **11a** than CIS. Moreover, **11a** (11 µmole/Kg) impeded CRC progression in C57BL/6 female mice xenografted human colon tumors by reducing VEGF mRNA expression (30%), GLUT1 mRNA expression (50%), vascularization (30%), and tumor size (90%) compared to OXA after 21 days [87]. Since **11a** showed better potency than OXA through blocking HIF-1 Pathway of CRC cells. Therefore, **11a** can be used in place of OXA in CRC treatment.

### 4.12. Anti-Metastasis Activity

Novel Ru(III)-based drug, Imidazolium-trans-tetrachloro(dimethylsulfoxide)imidazoleruthenium(III) or NAMI-A (**12a**) (Figure 6) was synthesized by Alessio et al. [163]. **12a** displayed anti-proliferative property against CRC cells by virtue of its anti-metastasis activity. Unlike other Ru-based drugs, **12a** focuses on the tumor microenvironment instead of showing direct cytotoxicity to the cell [164,165,166]. **12a** interact with actin-like proteins of the cell surface and collagens of the extracellular matrix and reduced invasive cancer cells mobility [167]. **12a** selectively targets surface adhesion receptor α5β1 integrin of colon cancer cells [88]. Highly invasive colon cell lines tend to express an increased level of α5β1 integrin [168], which is responsible for adhesion and migration of colon cancer cells through interacting with extracellular matrix proteins (ECM) [88,169,170]. According to Pelillo et al. [88], about 78% of cell adhesion is reduced by blocking α5β1 integrin site of HCT116 cells. **12a** blocked both the steps of adhesion and migration of the tumor cells by impairing the contact between α5β1 integrin and fibronectin of HCT116 cells in an inverse concentration-dependent manner (Figure 3→XII). **12a** (1–10 µM) inversely decreased the attachment of fibronectin with α5β1 integrin by 38–25%. **12a** (10–100 μM) also reduced the adhesion rate by 58–82%. Molecular insight of CRC revealed that **12a** altered the expression of the genes encoding the α5 and β1 subunit and, therefore, decreased the number of α5β1 integrin receptor molecules (Table 2). **12a** at a concentration of 1 μM downregulated α5 subunit encoding gene ITGA5 while 100 μM concentration upregulated ITGA5 expression up to 3.5-fold. Nonetheless, β1 subunit encoding gene ITGA1 did not respond to the alteration of concentrations [88]. Besides, the binding event activated autophosphorylation at the Tyr 397 site of the intracellular focal adhesion kinase (FAK) [170], which not only mediated tumor cell proliferation, survival, and migration [171] but also regulated the binding strength between integrins and ECM proteins [88]. **12a** (1–0 μM) decreased nearly 70–15% level of p-Tyr 397 FAK [88].

The metastasis of CRC cells is influenced by the hepatic microenvironment [170]. Bergamo et al. [169] reported that normal epithelial colon cells and hepatocytes release different soluble factors involved in the transcription of genes of the tumor cells associated with tumor growth, invasion, and migration. **12a** prevented transcription of those genes, thus inhibit the growth and dissemination of CRC cells. VEGF or MCP-1 either alone or in combination increased the migration ability of HCT116 cells. Exposure to **12a** decreased VEGF or MCP-1 induced migration of HCT116 cells [169]. As Pelillo et al. did not use any standard chemotherapeutics drug(s) (e.g., OXA, 5-FLU), further investigations are required to determine the potency of **12a** compared to standard drugs.

### 4.13. Lysosomal Dysfunction

Lysosomes contain various hydrolytic enzymes which degrade damaged proteins and organelles to regulate cellular functions [172]. However, releasing these hydrolase enzymes from lysosomes degrade other cytoplasmic organelles and lead to cell death [173]. Some Ru-complexes can be localized inside the lysosome, specifically where they cause lysosomal dysfunction [91,174,175]. Lysosomal dysfunction can be identified by unusual instigation of lysosomal enzymes, reduced lysosome-associated membrane proteins (LAMPs) expression as well as the permeability of lysosomes [176]. Disintegration of the lysosome induces the release of lysosomal hydrolases like cathepsin B from the lysosomal lumen to the cytosol, which makes the cells prone to lysosome-induced cell death (Figure 3→XIII) [173].

According to Xu et al. [91], half-sandwich Ru(II)-complexes bearing aryl- bis(imino)acenaphthene chelating ligands with fluorine group (**13a**) (Figure 6) induced lysosome mediated CRC cells death in vitro. **13a** displayed higher anti-proliferative activity compared to CIS against HT29 cells (5.91-fold), HCT116 cells (15.85-fold), and CT-26 cells (11.13-fold) in terms of IC_50_ values (Table 1). Moreover, **13a** suppressed tumor growth in CT-26 cells xenografted BALB/c mice model. Treatment of CT-26 cells with **13a** (0.575–2.3 µM) for 24 h elevated permeability of lysosomes and released cathepsin B [91]. **13a** downregulated LAMPs expression (20–60%) compared to control, thereby suppress metastasis [91,177]. Additionally, **13a** (0.575–1.15 µM) also increased ROS production (25–75%) in CT-26 cells [91]. Elevation of ROS level induced lipid peroxidation to rapture lysosome via destabilizing lysosomal membrane that leads to cell death [178]. 

### 4.14. Photodynamic Therapy

Photodynamic therapy (PDT) has emerged as a potential cancer therapy that is either used as a sole treatment or in combination with chemotherapy, surgery, and/or radiation [179]. PDT uses lights with appropriate wavelength to stimulate photosensitizer (PS) which mediates photochemical reaction to produce ROS and consequently kill tumors (Figure 6) [180].

Ru(II) PSs [Ru(2,2′-bipyridine)_2_(2-(2′,2″:5″,2′′′-terthiophene)-imidazo[4,5-f][1,10]phenanthroline)]_2_^+^ AKA TLD1411 (**14a**) and [Ru(4,4′-dimethyl-2,2′-bipyridine)_2_(2-(2′,2″:5″,2′′′-terthiophene)-imidazo[4,5-f][1,10]phenanthroline)]_2_^+^ AKA TLD1433 (**14b**) (Figure 7A) contain both photo-biological and photo-physical properties [92]. **14b** was originally synthesized by Sherri MacFarland and this Ru(II)-based photosensitizer entered clinical trial to treat bladder cancer through PDT. A phase I clinical study was conducted with **14b** (at 0.70 mgcm^−2^ dose) on six non-muscle-invasive bladder cancer patients (NCT03053635) and tumor relapse was not observed up to 180 days [181].

Ru(II) in **14a**, and **14b** makes the complexes specific towards cancer cells rather than normal cells and upon light irradiation increased singlet oxygen (^1^O_2_) quantum yield [182]. **14a** and **14b** induced fragmentation of DNA via photocleavage activities (Figure 7B) [92,183]. According to Fong et al. [92], **14a** (4 µM) and **14b** (1 µM) exhibited photodynamic effects which cause photon-mediated complete death of CT-26 cells. However, both **14a** (10 µM) and **14b** (10 µM) showed minimal toxicity (less than 10%) in the dark. The maximum tolerated dose was recorded to 36 and 103 mg/kg for **14a** and **14b**, respectively. Besides, in vivo treatment of **14a** and **14b** modulated tumor cell regression. Four hours post-intrathecal administration of **14a** (36 and 2 mg/kg), and **14b** (53 and 5 mg/kg) in BALB/c mice, followed by irradiation with a continuous wave or pulsed light sources (λ = 525–530 nm, H = 192 Jcm^−2^) for 30 min (with 30 s on/off cycle) exhibited a higher tumor growth reducing efficacy after 24 h. **14a** (2 mg/kg) and **14b** (5 mg/kg) delayed the tumor growth for 8 and 9 days, respectively. Both the compounds increased survival time in a dose-dependent manner. However, **14b** extended survival time by 5-fold compared to **14a** [92]. Treatment with **14b** also induced antitumor immunity in the colon cancer-containing mouse model [181]. Considering all observations, **14b** could be considered as a promising PDT agent in CRC treatment.

The Pt-based drug, Pt(II) 2,6-dipyrido-4-methyl-benzenechloride (PMB) also induced PDT-mediated DNA damage of CRC cells in a similar way [184]. Since **14b** and PMB were not investigated under similar experimental conditions, thus the potency of **14b** and PMB could not be compared.

## 5. Ru-Nanocomplexes in CRC Theranostics

### 5.1. CRC Diagnosis

RuNPs-based nanoformulations could facilitate the early diagnosis of CRC [30,185,186]. Xu et al. [30] constructed hollow mesoporous RuNPs (HMRuNPs) which are efficient in in vivo tumor imaging, drug loading, and combined treatment for CRC. Hollow mesoporous Ru containing fluorescent complex with anticancer activity (RBT) and bispecific antibodies (SS-Fc, anti-CD16, and anti-CEA) (**15a**) selectively accumulated into CRC cells by both active and passive targeting. Active targeting is mediated through antibody SS-Fc which can detect carcinoembryonic (CEA) antigen on CRC cell lines (i.e., HCT116, SW480, HT29, and Lovo) and attach with natural killer (NK) cells to induce an immune response. Passive targeting facilitates drug accumulation by the EPR effect. Moreover, the same compounds also have therapeutic effects. Treating BALB/c mice containing systemically administrated CT26-CEA tumor with **15a** (5 mg/kg) for every three days for a total of three treatments released RBT that generated ROS and appointed NK cells to initiate immune response, which in turn led to apoptosis and necrosis in CRC [30].

### 5.2. CRC Treatment

Conventional chemotherapy, as well as radiation therapy, conveys side effects and other limitations such as drug resistance [187]. The application of nanoformulations could overcome such shortcomings [188,189]. The advent of RuNPs offers excellent anticancer activity due to the high photothermal conversion rate, multiple oxidation states, and valence states [30]. Heffeter et al. [190] reported that micelle-like carriers (**MC-8b**) and nanoform of the established drug **8b** sidestepped the limitations of **8b** aqueous solutions undergo rapid hydrolysis to yield water-insoluble **8b** aqua complex, [mer,trans-[Ru(III)Cl_3_(Hind)_2_(H_2_O)]. Nevertheless, **MC-8b** (0.3 mg/mL **8b**) solutions were stable at 4°C for three months regarding precipitation. After 72 h of incubation, **MC-8b** was found to be more active than **8b** against HCT116 cells (3.91-fold) and Lovo cells (4.88-fold) in terms of their IC_50_ values. Moreover, **MC-8b** (IC_50_ = 41 µM) provided rapid onset of anticancer activity than **8b** (IC_50_ value of 135 μM) within only 1 h in HCT116 cells. Additionally, treatment of HCT116 cells with **MC-8b** (25 µM) exhibited higher apoptosis potential than **8b** by 6-fold. Western blotting indicated that **MC-8b** increased the expression of p53, phosphorylated P38, and JNK while decreased caspase-7 and PARP expression [190].

Besides, Zhu et al. [31] developed other Ru-based nanozymes, hollow Ru@CeO_2_ yolk-shell nanozymes in conjugation with antitumor drug Ru-complex (RBT) along with resveratrol (Res) and coated with DEPG (**16a**) which exhibited anti-metastasis and anti-tumor activity in orthotopic CRC through dual-chemotherapy/Photothermal therapy (PTT) with in situ oxygen supply [31]. Moreover, recurrence of more than 60% of post-surgical colorectal tumors is associated with the liver while more than 35% of all metastases occur solely in the liver [191,192]. **16a** with near infrared (NIR) efficiently inhibited intestinal, lung, and liver metastasis. **16a** contains antitumor Ru drugs, RBT, and Res, which exerted dual chemotherapeutic efficiency, while Ru@CeO_2_ holds efficient light-to-heat conversion potency. At the same time, **16a** catalyzed H_2_O_2_ to O_2_ in the tumor microenvironment (TME) and thereby overcame hypoxia by achieving in situ O_2_ supply and reduce HIF-1α hypoxic staining signal. Treatment with **16a** (5 mg/kg) for every three days for a total of three treatments overcame tumor hypoxia and obtained dual-chemotherapy/PTT in BALB/c mice bearing CT-26 cells. The excellent biocompatibility of the nanozyme is achieved due to the DPEG coating that prevented the occurrence of hemolysis even at a high dose concentration [31].

Like RuNPs, Pt-based nanoparticles (PtNPs) are also used in cancer diagnosis, drug delivery, treatment and can induce PTT-mediated apoptosis [193,194,195,196,197]. For example, Pt-based nanostructure, DPMNPs, was synthesized via coating dichloro(1,2-diaminocyclohexane)platinum(II) with poly[2-(*N*,*N*-dimethylamino)ethyl methacrylate]-poly(ε-caprolactone). DPMNPs exhibited a similar PTT-mediated anticancer effect in CRC cells [198].

## 6. Phase I Dose-Escalation Studies (Phase Ib Clinical Trials)

Dose escalation is an integral part of the phase I study which carefully looks for the optimal dose of a new drug to avoid therapeutic overdoses. In a dose-escalation study, the dosage of a drug is gradually increased until the side effects appear. Such a study is conducted on humans to assess the pharmacokinetics, pharmacodynamics, and safety of a new drug [199]. Rademaker-Lakhai et al. [200] conducted a phase I dose-escalation study with **12a** on seven CRC patients (*n* = 24, having solid tumors). **12a** at a dose concentration of 2.4–38.4 mg/m^2^/day caused no drug-induced toxicity. However, a dose concentration of 76.8–115 mg/m^2^/day resulted in causing diarrhea, phlebitis, and fatigue, while 400–500 mg/m^2^/day dose caused skin blisters lasting up to several months, which caused extreme pain. Considering all these data, the prescribed dose was set as 300 mg/m^2^ /day. However, this study did not provide effects of **12a** specifically on CRC patients rather gave a generalized overview of the effects of **12a** on 24 patients. Furthermore, in the phase I dose-escalation study **12a** stabilized disease for 21 weeks in a patient with lung cancer which prompted to organize a phase I/II trial on 32 non-small cell lung cancer patients in combination with gemcitabine [200,201]. Lentz et al. [202] performed a phase I dose-escalation study with **8b** on two CRC patients (*n* = 7, having various types of solid tumors). Intravenous administration of **8b** escalating from 25 to 600 mg (equivalent to 5–120.8 mg of Ru) twice weekly over 3 weeks caused no dose-limiting toxicity. Nonetheless, **8b** comes up with limitations regarding low solubility that makes it challenging to obtain proper dosage. Hence, its analogous sodium salt **8c** is largely used in clinical trials, which provides 35-fold better solubility [138]. In the study of Thompson et al. [203], 34 patients having solid tumors were treated with **8c** among whom 10 CRC patients were reported. **8c** (20–780 mg/m^2^) was infused in intravenous route on day 1, 8, and 15 of 28 days cycles. The maximum tolerated dose (MTD) was reported 625 mg/m^2^ with minor side effects [203,204]. However, none of the aforementioned phase I dose-escalation studies of **12a**, **8a**, and **8b** are listed on the clinicaltrial.gov website (www.clinicaltrial.gov) [205].

The only registered study was conducted by Burris et al. (trial registration number: NCT01415297) [206]. The phase Ib clinical trial was conducted to investigate MTD of **8c** (20–780 mg/m^2^) on 11 CRC patients (n = 46 patients having advanced solid tumors) and found similar MTD (625 mg/m^2^). However, ≥20% of the patients experienced adverse events that emerged from treatment which include fatigue, nausea, vomiting, dehydration, and anemia. In total, 59% of patients experienced ≤grade 2 and 37% of patients experienced grade 3 adverse effects though no patient was reported to have grade 4 adverse effects. It should be noted that both the studies did not present specific descriptions of the adverse effects on CRC patients [206].

Pt-based drugs, CIS, CAR, and picoplatin (PIC) have been used in combination with other chemotherapeutic drugs in clinical trials of CRC [205]. A Phase I clinical trial (NCT00465725) on various solid tumors, including CRC was studied using PIC only as an anti-proliferative agent [205]. A phase I clinical and pharmacological study with PIC revealed the MTD (150 mg/m^2^). Moderate level of anorexia, nausea, vomiting and transient metallic taste was evident and there was no significant sign of alopecia [207]. Peripheral neuropathy is the main disadvantage of OXA-based chemotherapy [208,209,210], whereas some Ru-based complexes (mentioned in this section) or PIC did not show such neurotoxic effects [211]. Therefore, Ru-based complexes or PIC could be used as an alternative to OXA in CRC patients with compromised neuronal function.

## 7. Toxicity of Ru-Drug Candidates

Pt-based drugs have toxic side effects which include neurotoxicity, nephrotoxicity, hepatotoxicity, ototoxicity, skin toxicity, myelosuppression, alopecia, diarrhea, fatigue, nausea, and vomiting [48,212,213]. Though Ru-based complexes exhibit parallel or better therapeutic efficacy than conventional Pt-based drugs, Ru-based complexes are promising for their lower toxicity [214]. [Ru(L-methionine)(2,2-bipyridine)(1,4-bis(diphenylphosphino)butane)]hexafluorophosphate (**17a**) and [Ru(l-tryptophan)(2,2-bipyridine)(1,4-bis(diphenylphosphino)butane)hexafluorophosphate (**17b**) displayed lower mutagenicity and genotoxicity in male Swiss mice compared to CIS at an equal concentration of 2 mg/kg body weight [215]. Furthermore, the in vivo biocompatibility of Ru-based drugs is another concern. **4b**, **4c**, **4d**, **17a**, and **17b** were reported to be safe at <300 mg/kg in vivo Swiss albino mice and Wistar rat models [82,83,84,215]. However, a therapeutic dose of >300 mg/kg elevated serum ALT, AST, ALP, BUN, creatinine, and glucose. At the same time, architectural alteration of kidney and liver was also evident [82,83,84]. Administering [Ru(Cl-terpyridine)(ethylenediamine)Cl][Cl] (**18a**) and [Ru(Cl-terpyridine)(1,2-diaminocyclohexane)Cl][Cl] (**18b**) at 2 mg/kg dose concentration in BALB/c mice bearing CT26 mouse colon carcinoma exhibited moderate histological changes in kidney, lung, and liver but no significant changes were found in heart architecture. However, there was a lack of changes in serum creatinine, urea, AST, and ALT level [62]. Furthermore, Ru-based drugs **12a** damaged kidneys by altering the glomeruli structure [216,217] and **8b** showed toxicity in bone marrow besides kidney [62].

Koch et al. [218] reported that some Ru-complexes (i.e., bipyridine, terpyridine, and phenanthroline Ru-complexes) acted as cholinesterase inhibitors in vitro and induced hind limb paralysis, respiratory distress, and death in respiratory failure as well as block curariform at the neuromuscular junction in in vivo mice model [218]. Furthermore, Ru is reported to inhibit Ca^2+^ uptake by mitochondria which possibly contributed to β-adrenergic and neuromuscular blocking [219,220,221]. Kruszyna et al. [221] described that some Ru-nitrosyl complexes at a concentration of (55–63 mg/kg) induced rapid death (after 10 min) while the rest of the complexes mediated death after 4-7 days a concentration ranging from 8.9 to 127 mg/kg. Ru is also retained in muscle and bone, rising concern about their long-term effect [221]. Thus, the toxicity of Ru-based drug candidates is a considerable issue before clinical applications. Ru-complexes including **1c**, **1d**, **4a**, **4f**, **6****a**, and **6****b**, displayed higher cytotoxic potential than OXA [27,77,87,115,116,117] while 5-FLU were found to be less potent than **4a** and **4f** in vitro and **3d** and **8b** in vivo [27,81,115,138]. Although the anticancer activity of Ru-complexes has been explored largely, their extended toxicity studies were not investigated. Therefore, preclinical chronic toxicity studies should be performed before considering these as potential drugs of CRC.

## 8. Conclusions

Ru-complexes displayed promising anticancer activities in the treatment of CRC. Ru(III) and Ru(II) were the most investigated oxidation states against all cancer cells including CRC cells, where the former one acts as a prodrug and converts to Ru(II) in the tumor microenvironment. Ru(II)-complexes displayed more reactivity compared to Ru(III) complexes. Many Ru-complexes were found to be more efficient than the Pt-based drugs; therefore, Ru could be used as an alternative to Pt-based drugs. Moreover, some Ru-complexes were reported to be more effective compared to the conventional chemotherapeutic drug (5-FLU) which has been used as first-line treatment against CRC. Though Ru conjugation with organic molecules could enhance anticancer activity through a synergistic effect [82,83,84], sometimes Ru-complexes are found to be less potent compared to parent organic molecules against CRC cells [90,222]. While conjugation of Ru-compounds with RuNps enhanced cellular uptake, selectivity, and drug delivery in CRC cells. Therefore, higher attention should be given to this field. Finally, extensive preclinical studies should be formed to confirm the efficacy, elucidating the potential mechanism of action(s), and toxicity of Ru-complexes or Ru-nanoformulations before considering them as potent drug candidates against CRC.

## Figures and Tables

**Figure 1 pharmaceutics-13-01295-f001:**
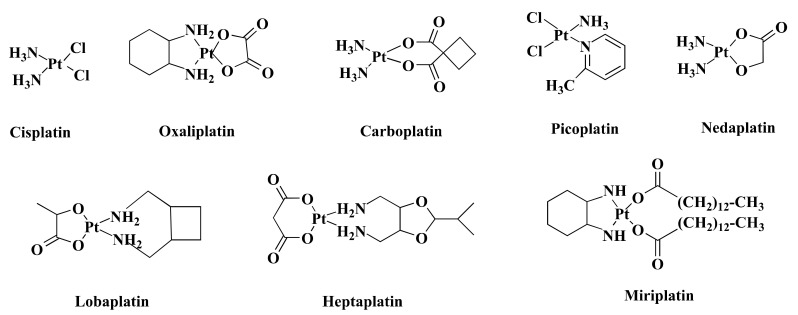
Chemical structure of some Pt-based drugs. Cisplatin, Oxaliplatin, Carboplatin, and Picoplatin have been used worldwide in cancer treatment. Besides, Nedaplatin, Lobaplatin, Heptaplatin, and Miriplatin have been using regionally. Among the Pt-based drugs, only Oxaliplatin is approved by FDA in CRC treatment.

**Figure 2 pharmaceutics-13-01295-f002:**
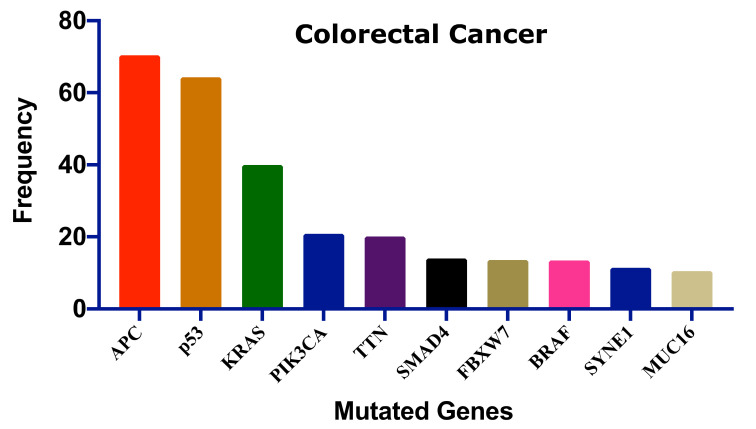
The ten most frequent mutations in colorectal cancer. This frequency distribution was calculated based on the cBioPortal data on 2322 CRC patients [46]. Adenomatous polyposis coli: APC, Tumor protein p53: p53, Kirsten rat sarcoma: KRAS, Phosphatidylinositol-4,5-Bisphosphate 3-Kinase catalytic subunit alpha: PIK3CA, Titin: TTN, SMAD family member 4: SMAD4, F-Box and WD repeat domain containing 7: FBXW7, B-Raf proto-oncogene: BRAF, Spectrin repeats containing nuclear envelope protein 1: SYNE1, Mucin 16: MUC16.

**Figure 3 pharmaceutics-13-01295-f003:**
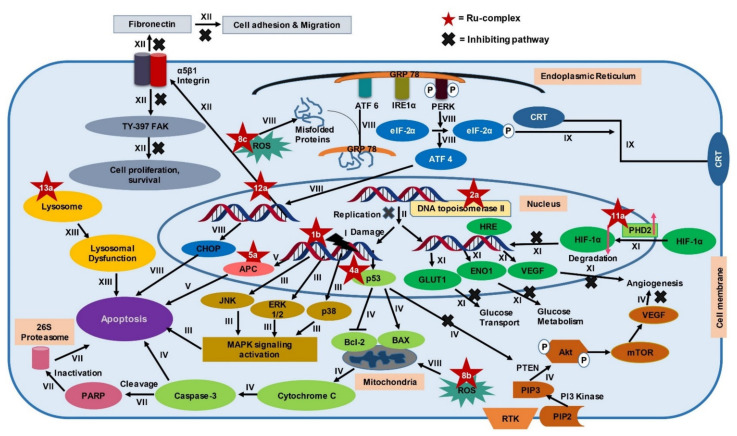
Summary of the suggested molecular mechanisms of some representative Ru-complexes. Different Ru-complexes interact with different cellular signaling pathways of colon cancer cells, thereby suppressing cell growth, proliferation, and migration. Some Ru-complexes can selectively accumulate into subcellular organelle in CRC cells and causes dysfunction leading to cell death. Tyrosine 397 focal adhesion kinase: Tyr-397 FAK; Reactive oxygen species: ROS; Glucose-regulated protein 78: GRP 78; Activating transcription factor 6: ATF6; Inositol-requiring protein 1α: IRE1α; PrKr-like ER kinase: PERK; Eukaryotic initiation factor2α: eIF2α; Calreticulin: CRT; C/eBP homologous protein: CHOP; Adenomatous polyposis coli: APC; Tumor protein P53: p53; B-cell lymphoma: Bcl-2; Bcl-2-associated X: BAX; c-Jun N-terminal kinases: JNK; Extracellular signal-regulated protein kinase 1 and 2: ERK1/2; Mitogen-activated protein kinases: MAPK; Receptor tyrosine kinases: RTK; Phosphatidylinositol 4,5-bisphosphate: PIP2; phosphatidylinositol (3,4,5)-trisphosphate: PIP3; Phosphatase and TENsin homolog deleted on chromosome 10: PTEN; Serine/threonine kinase 1: Akt1; Mammalian target of rapamycin: mTOR; Vascular endothelial growth factor: VEGF; Hypoxia-inducible factor-1 α: HIF-1α; Prolyl hydroxylase 2: PHD2; Hypoxia-responsive element: HRE; Glucose transporter 1: GLUT1; Alpha-enolase: ENO1.

**Figure 5 pharmaceutics-13-01295-f005:**
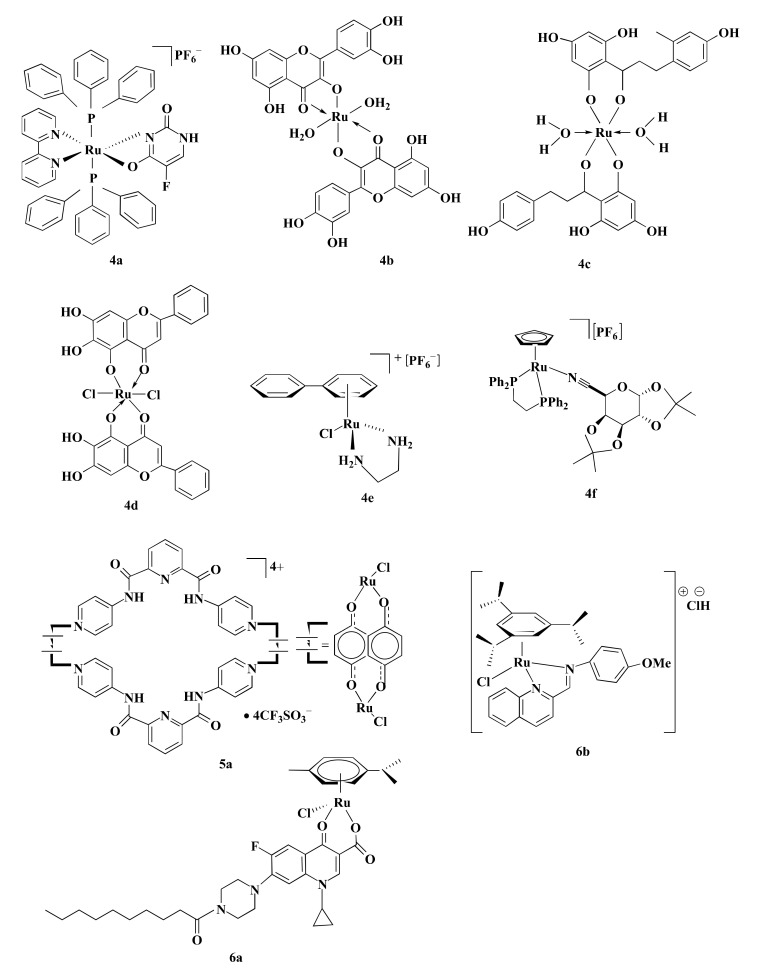
Chemical structures of some Ru-complexes that mediated CRC cell apoptosis through p53 dependent or independent way. Some Ru-complexes induced apoptosis of CRC cells via p53 dependent caspase-3 mediated signaling (**4**a**–f**), increasing APC and p53 gene expression (**5a**) and p53 independent activity (**6a**,**b**).

**Figure 7 pharmaceutics-13-01295-f007:**
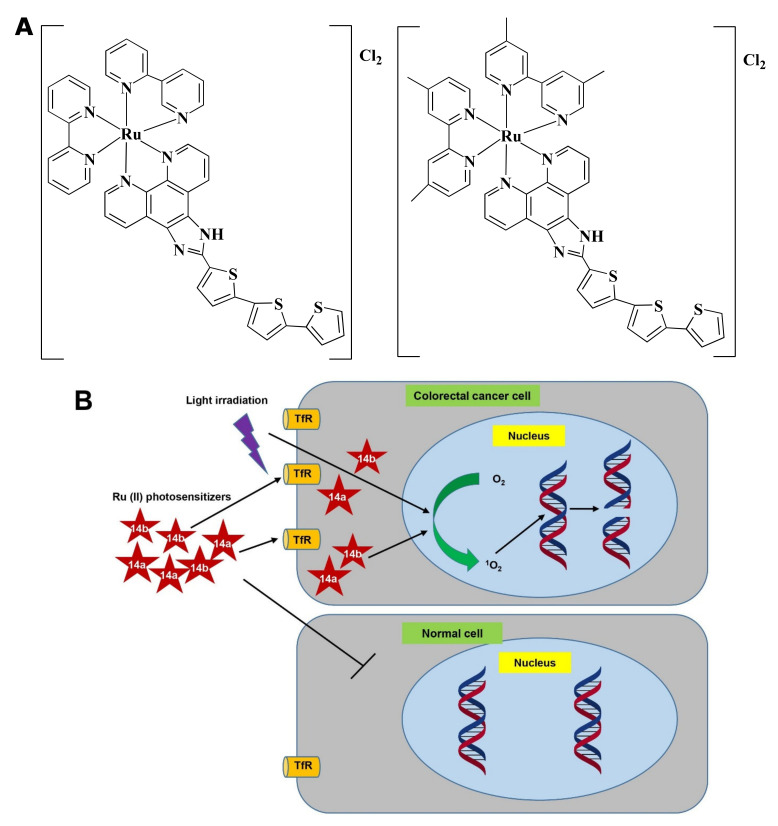
Photodynamic effects of Ru-complex in colorectal cancer and normal cells. (**A**) Chemical structure of **14a** and **14b**. (**B**) Ru-complexes having photosensitizing properties selectively kill CRC cells. Ru-complex with photosensitizing property can selectively enter cancer cells, and with the presence of light irradiation, O_2_ is converted into ^1^O_2_. ^1^O_2_ interact with DNA to mediate cell death via photocleavage activity. Tryptophan receptor: TfR, Oxygen: O_2_, Singlet oxygen: ^1^O_2_.

**Table 1 pharmaceutics-13-01295-t001:** Potency of Ru-based drug candidate(s) compared to conventional anticancer drugs.

Compounds or Drugs	Oxidation State	Assay Name	CRC Cell Lines and IC_50_ (µM)	Up-Regulated Protein	Down Regulated Protein	Cell Cycle Arrest	Corresponding Conventional Drugs IC_50_ (µM)	References
**1a**	III	TBE	HCT116 (>100) *	NR	NR	NR	HCT116 (CIS = 7.65 µM) *	[75]
**1b**	II	SRB	LS174T (7.7 µmol/L), LoVo (8.1 µmol/L) *****	NR	NR	NR	LS174T (CIS = 4.6), LoVo (CIS = 0.7) *****	[76]
**1c**, **1d**	II	CS	DLD1(**1c** = 10.2), (**1d** = 7.5) *	p53, p21, GAPDH	PARP	G2/M	DLD1 (OXA = 11.3) *	[77]
**2a**, **2b**	II	MTT	HCT116 (**2a** = 50.5, **2b** = 153), Caco-2 (**2a** = 26.3, **2b** = 121) ***	NR	NR	NR	HCT116 (CIS = 41.7, ETP = 18.3), Caco-2 (CIS = 14.9, ETP = 16.5) ***	[73]
**2c**	II	MTT	HCT116 (8.6), Caco-2 (6.6) ***	NR	NR	NR	HCT116 (ETP = 18.3), Caco-2 (ETP = 16.5) ***	[74]
**4a**	II	AB	HCT116 (1.5) ***	caspase-3	NR	ND	HCT116 (DOX = 0.5, OXA = 4.3, 5-FLU = 4.1)	[27]
**4f**	II	MTS	HCT116 (0.45) ***	Caspase-3, caspase-7	NR	NR	HCT116 (OXA = 0.45, 5-FLU = 3.80) ***	[115]
**5a**	II	MTT	HT-15 (6.9) *	p53, APC	NR	NR	HT-15 (CIS = 13.2, DOX = 15.9) *	[86]
**6a**	II	SRB	HCT116 (1.33) ***	NR	NR	Both S and G2/M phase	HCT116 (CIS = 5.1, OXA = 3.99) ***	[116]
**6b**	II	MTT	HCT116 (1.04), SW480 (7.3) ***	NR	NR	G0/G1 (2.5 µM) G2/M (10 µM)	HCT116 (OXA = 2.06), SW480 (OXA = 2.65) ***	[117]
**8a**	II	MTT	Caco-2 (6.16) **	NR	NR	G0/G1	Caco-2 (CIS = 17.9) **	[118]
**10a**	II	BP	HCT116 (5.22) ***	NR	NR	NR	CIS = No effect	[78]
**11a**	II	MTT	HCT116 (2.63) **	NAD^+^	HIF1α, VEGF GLUT1 ENO1	NR	HCT116 (CIS = 6.33) **	[87]
**13a**	II	MTT	HT29 (3.2), HCT116 (2.7), CT-26 (2.3) *	Cathepsin B	NF-kB p65, MMP-2, MMP-9, LAMP1	NR	HT29 (CIS = 18.9), HCT116 (CIS = 42.8), CT-26 (CIS = 25.6) *	[91]

trans-[RuCl_4_(1,3,5-triaza-7-phosphaadamantane protonated at one N atom)_2_]Cl: **1a**, Ru-5-chloro-3-((5-(3-(4-methyl-1,4-diazepane-1-carbonyl)phenyl)furan-2-yl)methylene)indolin-2-one: **1b**, [(η6-fluorene)RuII(ethylenediamine)Cl]^+^: **1c**, [(*η*6-9,10-dihydrophenanthrene)RuII(ethylenediamine)Cl]^+^: **1d**, [(η6-p-cymene)Ru(piperonal-N(4)-ethylthiosemicarbazone)Cl]Cl: **2a**, [(η6-p-cymene)Ru(piperonal-N(4)-phenylthiosemicarbazone)Cl]Cl: **2b**, [(1,10-phenanthroline)2Ru(2-(benzo[d][1,3]dioxol-5-ylmethylene)-*N*-methylhydrazinecarbothioamide)](hexafluorophosphate)2: **2c**, [Ru(piplartine)(1,1 bis(diphenylphosphino)ferrocene)(2,2′-bipyridine)](hexafluorophosphate)2: **3b**, [Ru(piplartine)(1,4-bis(diphenylphosphino)butane)(2,2′-bipyridine)](hexafluorophosphate)2: **3c**, Ru(II) thymin complex: **3d**, [Ru(5-Fluorouracil)(triphenylphosphine)_2_(2,2′-bipyridine)]hexafluorophosphate: **4a**, [(η^5^-C_5_H_5_)Ru(1,2-bis(diphenylphosphino)ethane)]^+^ bearing the galactose nitrile derivative ligands: **4f**, [Ru_4_(p-cymene)_4_(5,8-dioxydo-1,4-naphthaquinonato-)_2_(2,6-bis(N-(4-pyridyl)carbamoyl)pyridine)_2_][4CF_3_SO_3_]: **5a**, [Ru(η^6^-*p*-cymene)( 7-(4-(Decanoyl)piperazin-1-yl)-ciprofloxacin_-H_)Cl]: **6a**, [(η6-1,3,5-triisopropylbenzene)RuCl(4-methoxy-N (2-quinolinylmethylene)aniline)]Cl: **6b**, [(η6-p-cymene)RuCl[118]]: **8a**, trans-[tetrachloro-bis(1H-indazole)ruthenate(III)]: **8b**, Sodium trans-[tetrachloride-bis(1H-indazole)ruthenate(III)]**: 8c**, [Ru(p-cymene)Cl_2_(µ-1,1-bis(diphenylphosphino)methane)Au(IMes)]-ClO_4_: **10a**, Ruthenium derived compound 11: **11a**, Half-sandwich Ru(II) complexes bearing aryl-BIAN chelating ligands: **13a,** Cisplatin: CIS, Oxaliplatin: OXA, Doxorubicin: DOX, 5-Fluorouracil: 5-FLU, Etoposide: ETP**;** The half maximal inhibitory concentration: IC_50_, Not reported: NR, Not detected: ND, Trypan blue exclusion: TBE, Sulforhodamine B: SRB, Clonogenic survival: CS, 3-(4,5-dimethylthiazol-2-yl)-2,5-diphenyl tetrazolium bromide: MTT, Alamar blue: AB, 3-(4,5-dimethylthiazol-2-yl)-5-(3-carboxymethoxyphenyl)-2-(4-sulfophenyl)-2H-tetrazolium): MTS, Neutral red uptake: NRU, BluePresto^TM^: BP, Tumor protein P53: p53, Cyclin-dependent kinase inhibitor 1: p21, Glyceraldehyde 3 phosphate dehydrogenase: GAPDH, Poly (ADP-ribose) polymerase: PARP, Phosphorylated c-Jun N-terminal kinases 2: p-JNK 2, Extracellular signal-regulated kinases 1: ERK1, Mitogen-activated protein kinases α: p38α, H2A histone family member X: H2AX, Adenomatous polyposis coli: APC, Matrix metalloproteinase-2: MMP-2, B-cell lymphoma 2: Bcl-2, Nicotinamide adenine dinucleotide: NAD^+^, Hypoxia inducible factor-1 α: HIF-1α, Vascular endothelial growth factor: VEGF, Glucose transporter 1: GLUT1, Enolase 1: ENO1, Nuclear factor of the κ-chain in B-cells p65: NF_k_B p65, Matrix metalloproteinase-2: MMP-2, Matrix metalloproteinase-9: MMP-9, Lysosomal associated membrane protein 1: LAMP1,. References: Ref, After 24 h: *, After 48 h: **, After 72 h: ***, After 6 days: *****.

**Table 2 pharmaceutics-13-01295-t002:** Cytotoxic potential of some Ru-complexes towards CRC cells.

Compounds or Drugs	Oxidation State	Assay Name	CRC Cell Lines and IC_50_ (µM)	Up-Regulated Protein	Down Regulated Protein	Cell Cycle Arrest	References
**3a**	II	NR	HCT116	p21, Bad, p-p38 MAPK, ATF2, Stat1, MMP	Bax	G1	[79]
**4b**	II	MTT	HT29 (<100) **	p53, caspase-3, Bax	Akt1, mTOR, VEGF, Bcl-2, PCNA, WNT, β-catenin	G0/G1	[83]
**4c**	II	MTT	HT29 (>100) *	p53, caspase-3, Bax	Akt1, p-Akt, mTOR, p-mTOR, VEGF, Bcl-2, NF-κΒ, MMP-9, PCNA	G0/G1	[82]
**4d**	II	MTT	HT29 (~30) **	p53, caspase-3, Bax	Akt1, mTOR, VEGF, Bcl-2, PCNA, WNT, β-catenin	G0/G1	[84]
**4e**	II	SRB	HCT116 (8) HCT116 p53 (16) ****	p53, p21/WAF, Bax	NR	G1 and G2	[85]
**7a, 7b, 7c**	II	MTT	HCT116 (NR)	p53, caspase-3	PARP	NR	[90]
**8b**	III	NRU	SW480 (30), LT97 (50) *	Caspase-3	PARP, MMP, Bcl-2	NR	[119]
**8c**	III	MTT	HCT116 (20), SW480 (40) *	eIF2α, ATF4, CHOP	NR	NR	[89]
**8c**	III	NR	HT15, HCT116, HT29 (NR)	eIF2α, CRT, HMGB-1, ATP, Beclin-1, LC3A/B-II	NR	NR	[120]
**12a**	III	MTT	HCT116	RND-1, SIK-1	α5β1 integrin, VEGF, MCP-1	NR	[88]
**14a**, **14b**	II	PBCV	CT-26 (NR)	NR	NR	NR	[92]

Ru(II) naphthalimide *N*-Heterocyclic Carbene compounds: **3a**, Ru-Quercetin: **4b**, Ru-Phloretin: **4c**, Ru-Baicalein: **4d**, [Ru(biphenyl)Cl(1,2-ethylenediamine)]+ with hexafluorophosphate: **4e**, [Ru(p-cymene)(curcumin)Cl]: **7a**, [(Benzene)Ru(curcumin)Cl]: **7b**, [Ru(hexamethylbenzene)(curcumin)Cl]: **7c**, trans-[tetrachloro-bis(1H-indazole)ruthenate(III)]: **8b**, Sodium trans-[tetrachloride-bis(1H-indazole)ruthenate(III)]**: 8c****,** [Imidazolium-trans-tetrachloro(dimethylsulfoxide)imidazoleruthenium(III)]: **12a**, [Ru(2,2′-biyridine)_2_(2-(2′,2″:5″,2′′′-terthiophene)-imidazo[4,5-f][1,10]phenanthroline)]_2_^+^: **14a**, [Ru(4,4′-dimethyl-2,2′-bipyridine)_2_(2-(2′,2″:5″,2′′′-terthiophene)- imidazo[4,5-f][1,10] phenanthroline)]_2_ ^+^: **14b**. 3-(4,5-dimethylthiazol-2-yl)-2,5-diphenyl tetrazolium bromide: MTT, Sulforhodamine B: SRB, Neutral red uptake: NRU, Presto Blue Cell Viability: PBCV, Cyclin-dependent kinase inhibitor 1: p21, Bcl-2 associated agonist of cell death: Bad, BCl-2 associated X: BAX, Tumor protein P53: p53, Phosphorylated p38 MAP Kinase: p-p38 MAPK, Activating transcription factor 2: ATF2, Signal transducer and activator of transcription 1: Stat1, Matrix metallo proteinases: MMP, Serine/threonine kinase 1: Akt1; Mammalian target of rapamycin: mTOR; Vascular endothelial growth factor: VEGF, B-cell lymphoma 2: Bcl-2, Proliferating cell nuclear antigen: PCNA, Wingless-related integration site: WNT, Nuclear factor of the κ-chain in B-cells: NF_k_B, Matrix metalloproteinase 9: MMP-9, Poly (ADP-ribose) polymerase: PARP, Eukaryotic translation initiation factor 2α: eIF2α, Activating transcription factor 4: ATF4, C/eBP homologous protein: CHOP, Calreticulin: CRT, Adenosine triphosphate: ATP, High mobility group protein B1: HMGB-1, Rho Family GTPase 1: RND1, Salt inducible kinase 1: SIK-1, Monocyte chemotactic protein-1: MCP-1. After 24 h: *, After 48 h: **, After 96 h: ****.

## Data Availability

Data reported in Figure 2 are available at cBioportal database (https://www.cbioportal.org/) (accessed on 15 July 2021).

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
