# Peer review of "Ruthenium Complexes: An Alternative to Platinum Drugs in Colorectal Cancer Treatment"

_pharmaceutics, 2021, doi:10.3390/pharmaceutics13081295_

Round 1

Reviewer 1 Report

This is a very nice manuscript giving an excellent overview of recent results in the field of ruthenium complexes as alternatives of platinum drugs in 
colorectal cancer treatment. I can recommend this contribution for publication in Pharmaceutics.

Author Response

Response to Reviewer 1 Comments

Point 1: This is a very nice manuscript giving an excellent overview of recent results in the field of ruthenium complexes as alternatives of platinum drugs in

colorectal cancer treatment. I can recommend this contribution for publication in Pharmaceutics.

Response 1: Many thanks for your positive comment and recommendation. We hope that the revised manuscript is much better now. Thank you

Reviewer 2 Report

Mahmud et al. present a complete and exhaustive overlook on the ruthenium complexes as possible anticancer drugs in the CRC treatment examining several aspects such as the different mechanism of actions, the cytotoxic studies and the clinical trials, the nanoformulations to improve the drugs efficiency.

The review is well written and comprehensive, a hard work has been done on data collection, however, according to my point of view, a fine-tuning of the manuscript is needed to make it more reader friendly.

Specifically, in the paragraph 5, the different mechanisms of action are clear and well highlighted in each subsection but is not clear which classes of ruthenium complexes are referred to. I suggest indicating at the beginning of each subparagraph the classe(s) of the involved ruthenium complexes with the referred figure ( e.g. 5.1. Some Ru-complexes (Figure 3) such as Ru(III)-PTA…)

Minor comments:

  • Figure 3, 4, 5 are not indicate in the text.
  • Line 136 “Ru-complex can exist in several oxidation states…” is redundant, already written at line 126.
  • Line 147. The sentence “mechanism contains controversy…” is unclear, please modify
  • Line 158. The sentence “investigate their promise in CRC treatment…” is unclear, please modify
  • In the paragraph 5.4 some proteins up and downregulated have been indicate (line 246-263). Please add here a sentence to explain the used methodology to identify these proteins and not in the middle of the subparagraph (line 293 “Western blot results…)

Author Response

Response to Reviewer 2 Comments

Point: Mahmud et al. present a complete and exhaustive overlook on the ruthenium complexes as possible anticancer drugs in the CRC treatment examining several aspects such as the different mechanism of actions, the cytotoxic studies and the clinical trials, the nanoformulations to improve the drugs efficiency. The review is well written and comprehensive, a hard work has been done on data collection, however, according to my point of view, a fine-tuning of the manuscript is needed to make it more reader friendly.

Response: We appreciate your optimistic feedback.

Point 1: Specifically, in the paragraph 5, the different mechanisms of action are clear and well highlighted in each subsection but is not clear which classes of ruthenium complexes are referred to. I suggest indicating at the beginning of each subparagraph the classe(s) of the involved ruthenium complexes with the referred figure ( e.g. 5.1. Some Ru-complexes (Figure 3) such as Ru(III)-PTA…)

Response 1: Many thanks for your comments and suggestions. In the revised manuscript, we have added precise figure numbers alongside name of the Ru-complexes. We have also corrected the figure legends (Figure 4, 5, and 6) to better reflect the class of Ru complexes based on the mechanisms. In addition, we have modified Figure 3 (previously Figure 2) to highlight the classes of Ru-complexes. We hope it’s alright now. Thank you

Point 2: Figure 3, 4, 5 are not indicate in the text.

Response 2: We apologize for this error. In the revised manuscript, we have added figure numbers in the text. Thank you

Point 3: Line 136 “Ru-complex can exist in several oxidation states…” is redundant, already written at line 126.

Response 3: Thank you for your comment. We agree and we have removed the redundant information from the revised manuscript. Thank you

Point 4: Line 147. The sentence “mechanism contains controversy…” is unclear, please modify.

Response 4: In the revised manuscript, we have modified the sentence (Line 154-156). Thank you

Point 5: Line 158. The sentence “investigate their promise in CRC treatment…” is unclear, please modify.

Response 5: In the revised manuscript, we have modified the sentence (Line 166-167). Thank you for your comment.

Point 6: In the paragraph 5.4 some proteins up and downregulated have been indicate (line 246-263). Please add here a sentence to explain the used methodology to identify these proteins and not in the middle of the subparagraph (line 293 “Western blot results…)

Response 6: Many thanks for your comment. We have added a sentence to explain the used methodology to identify these proteins (Line 301-308). Thank you

Reviewer 3 Report

The manuscript by Islam et al. collects interesting information about Ruthenium complexes as Colorectal cancer drugs. The field of Metal in medicine is critical as an alternative therapy to organic drugs and the review collect the most important examples. This work could help to understand the mechanism of action of the complexes and could stablish an interesting SAR’s. However, the work fails in several important aspects which under my point of view are not difficult to mend but mandatory to address. Those are summarized as follows:

First, I think the title does not reflect the content of the review. If they try to see Ru complexes as an alternative for platinum drugs, the authors should discuss the differences in between Ru and Pt found in mechanism (5) nanoparticules formulation (6) and clinical trials (where only Pt complexes have gone through). That will be a fantastic review, but extremely difficult not to say impossible to perform as many of the research published with platinum is not available for the Ru complexes. Not to mention the huge number of papers in nano formulation with platinum, and the little knowledge of Ru limitations as it has not gone through advance clinical trials yet and those data are not available. So, I recommend joining section 2 to 3, as section 3 explained the difficulty to tackle Colorectal cancer and ends nicely the section. The authors should modulate the title in accordance.

Second, the authors must extend and clarify section 5. The mechanism of action section is the highlight of the paper, and though the election of the sections is adequate the order is hectic and some of them are not completed.

For example, 5.1. DNA damage is clear enough with an analysis of the interaction of the Ru complexes, and no covalent binding has been reported. The order continues with 5.2. and the topoisomerase II analysis that follows a logic analysis based on the role on the DNA replication and unwinding, but after, the authors start the intrinsic signalling of apoptosis: MAPK signalling, p53 dependence to jump to the proteasome and back to p53 not dependent pathways with no reasoning.

The sections 5.3 to 5.12 show no conclusions, but a description of the values achieved in the experiments of the papers. The authors should improve those sections with final sentences clarifying the contribution of the analysis and linking with figure 2 (might be colouring the arrows by section within the paper)

NAMI A is a Ru drug which has been broadly studied as it is the most effective versus methastasis. I found section 5.12 particularly focused as a list of values, and I encourage the authors to read Enzo alessio’s personal input (father of NAMI A)  https://doi.org/10.1002/ejic.201600986 and include this to the section.

Speaking of missing citations, I must say that the authors need to CAREFULLY revise the citations and cite properly the first authors who published the compounds instead of a review or a citation of a citation. Section 5.14 studied McFarland compounds and the authors has not mentioned at all the original work. I not only encourage but urge the authors to study and include the original papers and recent review from the laboratory of the Prof Mcfarland. Sherry MacFarland synthesized the thiophen Ru derivatives and took them to the most advanced clinical trials step in Ru history. The review will allow the authors to include the structure activity relationship of the series and correlate those with their photodamaging properties. https://doi.org/10.1021/acs.chemrev.8b00211.

In general the references on this review need to be revised and include the first publications (original papers) though afterwards other citation were used.

Author Response

Response to Reviewer 3 Comments

Point: The manuscript by Islam et al. collects interesting information about Ruthenium complexes as Colorectal cancer drugs. The field of Metal in medicine is critical as an alternative therapy to organic drugs and the review collect the most important examples. This work could help to understand the mechanism of action of the complexes and could stablish an interesting SAR’s. However, the work fails in several important aspects which under my point of view are not difficult to mend but mandatory to address. Those are summarized as follows:

Response: We appreciate your insightful comments. Thank you

Point 1: First, I think the title does not reflect the content of the review. If they try to see Ru complexes as an alternative for platinum drugs, the authors should discuss the differences in between Ru and Pt found in mechanism (5) nanoparticules formulation (6) and clinical trials (where only Pt complexes have gone through). That will be a fantastic review, but extremely difficult not to say impossible to perform as many of the research published with platinum is not available for the Ru complexes. Not to mention the huge number of papers in nano formulation with platinum, and the little knowledge of Ru limitations as it has not gone through advance clinical trials yet and those data are not available. So, I recommend joining section 2 to 3, as section 3 explained the difficulty to tackle Colorectal cancer and ends nicely the section. The authors should modulate the title in accordance.

Response 1: Thank you for your comment. In the revised version we have added the mechanism of platinum drugs where applicable. For instance, in section 4.1 the mechanism of oxaliplatin has been added (Line number 212-215).

In the section, 4.14 and 5.2 we have added relative information of platinum complex and nanoparticles (Line number 627-630 and 691-695).

In the clinical trial selection, we have added information on the Pt-based drug candidates for CRC treatment. We hope the added information of these sections enriches this review and now the title fit with the content of the review (Line number 732-741).

We have also merged sections 2 and 3 according to your suggestion as the section ends nicely and added a figure containing the chemical structure of some Pt-based drugs (Figure 1).

We hope the revised parts and rearrangements are alright now. Thank you

Point 2: The authors must extend and clarify section 5. The mechanism of action section is the

highlight of the paper, and though the election of the sections is adequate the order is hectic and some of them are not completed.

For example, 5.1. DNA damage is clear enough with an analysis of the interaction of the Ru

complexes, and no covalent binding has been reported. The order continues with 5.2. and the

topoisomerase II analysis that follows a logic analysis based on the role on the DNA replication and unwinding, but after, the authors start the intrinsic signalling of apoptosis: MAPK signalling, p53 dependence to jump to the proteasome and back to p53 not dependent pathways with no reasoning.

Response 2: We have rearranged the order sequence in section 4 (previously 5) in the revised manuscript according to your valuable comment. p53 linked mechanisms were placed in sequential order and hope the current version of the manuscript reads well. We hope the new rearrangement is alright now. Thank you

Point 3: The sections 5.3 to 5.12 show no conclusions, but a description of the values achieved in the experiments of the papers. The authors should improve those sections with final sentences clarifying the contribution of the analysis and linking with figure 2 (might be colouring the arrows by section within the paper)

Response 3: We have added concluding sentences in sections 4.3 to 4.12 (Previously 5.3 to 5.12) (Line number 258-259, 327-329, 365-368, 382-384, 402-405, 457-460, 501-503, 510-511, 535-537, 570-572) in the revised manuscript. We hope it’s alright now. Thank you

Point 4: NAMI A is a Ru drug which has been broadly studied as it is the most effective versus methastasis. I found section 5.12 particularly focused as a list of values, and I encourage the authors to read Enzo alessio’s personal input (father of NAMI A)  https://doi.org/10.1002/ejic.201600986 and include this to the section.

Response 4: We have revised the section and included the mentioned review article (Line number 543-545). Thank you for your suggestion.

Point 5: Speaking of missing citations, I must say that the authors need to CAREFULLY revise the citations and cite properly the first authors who published the compounds instead of a review or a citation of a citation. Section 5.14 studied McFarland compounds and the authors has not mentioned at all the original work. I not only encourage but urge the authors to study and include the original papers and recent review from the laboratory of the Prof Mcfarland. Sherry MacFarland synthesized the thiophen Ru derivatives and took them to the most advanced clinical trials step in Ru history. The review will allow the authors to include the structure activity relationship of the series and correlate those with their photodamaging properties. https://doi.org/10.1021/acs.chemrev.8b00211..

Response 5: We have revised section 4.14 (previously 5.14) and included the mentioned article (605-609). We also have replaced review articles with original research articles (Line 156, 543) and added some research papers which contain the synthesis of some compounds (Line 434, 521, 540). We hope it goes well now. Thank you
